# Superior thermoelasticity and shape-memory nanopores in a porous supramolecular organic framework

You-Gui Huang[1,2], Yoshihito Shiota[1], Ming-Yan Wu[3], Sheng-Qun Su[1], Zi-Shuo Yao[1], Soonchul Kang[1], Shinji Kanegawa[1], Guo-Ling Li[1], Shu-Qi Wu[1], Takashi Kamachi[1], Kazunari Yoshizawa[1], Katsuhiko Ariga[2], Mao-Chun Hong[3] & Osamu Sato[1]

Flexible porous materials generally switch their structures in response to guest removal or incorporation. However, the design of porous materials with empty shape-switchable pores remains a formidable challenge. Here, we demonstrate that the structural transition between an empty orthorhombic phase and an empty tetragonal phase in a flexible porous dodecatuple intercatenated supramolecular organic framework can be controlled cooperatively through guest incorporation and thermal treatment, thus inducing empty shape-memory nanopores. Moreover, the empty orthorhombic phase was observed to exhibit superior thermoelasticity, and the molecular-scale structural mobility could be transmitted to a macroscopic crystal shape change. The driving force of the shape-memory behaviour was elucidated in terms of potential energy. These two interconvertible empty phases with different pore shapes, that is, the orthorhombic phase with rectangular pores and the tetragonal phase with square pores, completely reject or weakly adsorb $N_2$ at 77 K, respectively.

[1] Institute for Materials Chemistry and Engineering, Kyushu University, 744 Motooka Nishi-ku, Fukuoka 819-0395, Japan. [2] World Premier International (WPI) Center for Materials Nanoarchitectonics (MANA), National Institute for Materials Science (NIMS), 1-1 Namiki, Tsukuba, Ibaraki 305-0044, Japan. [3] State Key Laboratory of Structure Chemistry, Fujian Institute of Research on the Structure of Matter, Chinese Academy of Science, Fuzhou, Fujian 350002, China. Correspondence and requests for materials should be addressed to M.-C.H (email: hmc@fjirsm.ac.cn) or to O.S. (email: sato@cm.kyushu-u.ac.jp).

Shape-memory materials are a class of dynamic materials capable of converting heat into mechanical strain (or vice versa)[1–5]. They change their morphological appearance upon application of an external stimulus, hold their temporary shape after the stimulus has been removed, and recover to their original morphology in the presence of a second external stimulus. For example, a polymer can exhibit a thermoresponsive shape-memory effect if the deformed shape induced by a programming process can be maintained temporarily and recover to its permanent shape upon heating[6,7].

Shape-memory effects are primarily observed in metal alloys[4,8], polymers[1,2] and ceramics[3]. The structural transition in some flexible microporous materials has been demonstrated to be reversibly initiated by both guest molecules and temperature[9,10]. Achieving empty shape-memory pores requires that the reversibility of the structural transition of flexible porous materials be achieved through cooperation of two different external stimuli, not just one stimulus. Recently, interconvertible empty phases have been realized in some microporous metal–organic frameworks (MOFs)[9–11] and molecular crystals either by chemical stimuli (mainly change of guest) or physical stimuli (for example, photoirradiation or temperature change)[12,13]. Empty shape-memory pores have also been achieved in a flexible porous coordination polymer through downsizing of the crystals, which suppresses structural mobility during desolvation and stabilizes a metastable open dried phase[14].

Our work has been focused on porous hydrogen-bonded organic frameworks (HOFs) because of the soft nature of their intermolecular interactions. In contrast to the explosive development of porous MOFs[15], the development of porous HOFs has lagged because the weak intermolecular interactions within HOFs are typically not sufficiently strong to establish their permanent porosities. Very recently, a few HOFs have been demonstrated to possess permanent porosities[16–18], and two exceptional examples[19,20] exhibit guest-induced structural changes. Herein, we demonstrate the empty shape-memory pores in a microporous supramolecular organic framework (SOF), wherein thermal treatment and guest incorporation cooperatively control the structural transition via a ferroelastic transition that involves pyridyl ring-flipping, leading to switchable sorption behaviour. The driving force of the shape-memory behaviour was elucidated in terms of potential energy. Moreover, superior thermoelasticity was observed in an empty orthorhombic phase, and the molecular-scale structural mobility could be transmitted to a macroscopic crystal shape change.

## Results

**Synthesis and structure of the supramolecular framework.** Biphenyl-3,3′,5,5′-tetracarboxylic acid (H₄BPTC) and 1,2-bis(4-pyridyl)ethane (BPE) were chosen to build porous binary HOFs because of their intrinsic flexibilities. Colourless needle-shaped crystals of $[(H_4BPTC) \cdot (BPE)_2] \cdot 0.33DMF$ (compound $1o \cdot 0.33DMF$) were readily synthesized by solvothermal reaction of $H_4BPTC$ and BPE in DMF at $120\,°C$. The formula was determined by single-crystal X-ray diffraction and thermogravimetric (TG) analyses; the IR spectrum of $1o \cdot 0.33DMF$ is shown in Supplementary Fig. 1. Single-crystal X-ray crystallography at $123\,K$ revealed that compound $1o \cdot 0.33DMF$ crystallizes in space group $Ccca$ (Supplementary Table 1) and consists of a dodecatuple intercatenated hydrogen-bonded framework with colossal adamantine cages (Fig. 1; Supplementary Fig. 2). Each $H_4BPTC$ is connected to its four neighbours by BPE through two pairs of hydrogen bonds, forming a chiral three-dimensional diamond (dia) supramolecular network with channels along all three crystallographic axes, including two types (large and

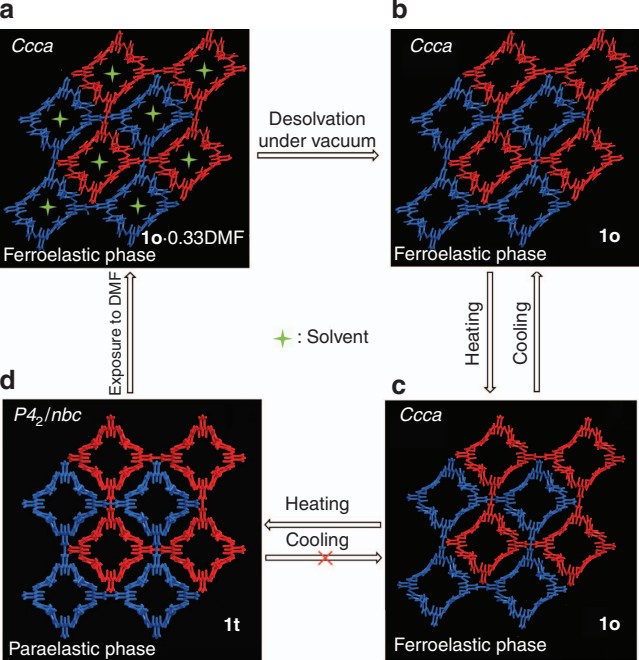

**Figure 1 | Schematic of the shape-memory effect in the flexible porous HOF.** (**a**) Structure of as-prepared compound $1o \cdot 0.33DMF$ at 123 K; (**b**) structure of compound $1o$ at 123 K; (**c**) structure of compound $1o$ at 393 K; and (**d**) structure of compound $1t$ at 438 K (red and blue frameworks represent the pair of sextuple interlocked frameworks with opposite handedness). To understand this cycle more easily, we can set compound $1t$ as the initial state. Guest incorporation induces ferroelastic transition, during which the host–guest interactions act as stress, inducing the structural strain and resulting in the transition of compound $1t$ to compound $1o \cdot 0.33DMF$. At lower temperatures, compound $1o$ is energetically preferred; therefore, the strain can be preserved after the stress is removed and the framework remains unchanged upon desolvation under vacuum. However, compound $1t$ can be recovered under thermal treatment via a reverse ferroelastic transition (transition from a ferroelastic phase to a paraelastic phase); therefore, shape-memory behaviour is observed.

small) of helical channels along the $c$ axis. The dia network consists of colossal adamantine cages assembled by four helicates (Supplementary Fig. 2a). Each adamantine cage comprises 10 $H_4BPTC$ and 12 BPE held together by 36 hydrogen bonds. The adamantine cage has a huge cavity with $5.9 \times 3.5\,nm^2$ windows (Supplementary Fig. 2c). Because of the large void space in a single framework, six identical dia networks interpenetrate in parallel, forming a sextuple intercatenated framework (Supplementary Fig. 2b), a pair of which with opposite handedness further interlocks at an interlocking angle $\omega$ of $72.3°$. This arrangement results in the channels along the $a$ and $b$ axes being fully blocked and the large channels along the $c$ axis being filled with small channels, leaving only alternatively arranging left-handed and right-handed small helical channels with dimensions of $3.2 \times 4.0\,Å^2$ in the framework (Fig. 1; Supplementary Fig. 3). Compound $1o \cdot 0.33DMF$ has a void space of $1,005\,Å^3$ per unit cell, which is 13% of the unit-cell volume. Such extrinsic porosity (that is, porosity resulting solely from the solid-state molecular packing) in molecular crystals is uncommon and challenging to design[12,16–20].

**Thermally induced phase transition.** TG analysis revealed that the solvent in the pores of compound $1o \cdot 0.33DMF$ represents $\sim 6.8\%$ of the total weight. The solvent can be removed by heating at $60\,°C$ under vacuum for 12 h, generating an empty

orthorhombic phase [(H$_4$BPTC)·(BPE)$_2$] (compound **1o**). TG, $^1$H NMR, and elemental analyses of the activated sample revealed that the solvent was completely removed (Supplementary Figs 4 and 5). Structural determinations were performed on a single crystal of compound **1o** heated from 123 to 438 K and then cooled back to 123 K (Supplementary Table 2). The single-crystal analysis shows that the framework remains almost unchanged upon desolvation. Furthermore, the framework of compound **1o** gradually breathed with increasing temperature and, at temperatures greater than 438 K, underwent a phase transition via pyridyl ring-flipping from an orthorhombic phase (compound **1o**, space group *Ccca*) to a tetragonal phase (compound **1t**, space group *P4$_2$/nbc*). The orientation relationship between the tetragonal phase and the orthorhombic phase can be described as

$$\begin{pmatrix} a_o \\ b_o \\ c_o \end{pmatrix} = \begin{pmatrix} 1 & 1 & 0 \\ -1 & 1 & 0 \\ 0 & 0 & 1 \end{pmatrix} \times \begin{pmatrix} a_t \\ b_t \\ c_t \end{pmatrix}$$ (refs 21,22) where $a_o$, $b_o$ and

$c_o$ are the lattice axes for the orthorhombic phase, and $a_t$, $b_t$ and $c_t$ are those for the tetragonal phase. In the following description, all the structures will be presented in the lattice of the orthorhombic phase. From 123 to 438 K, the *a* axis expanded by 14%, whereas *c* axis length remained almost constant. The length of *b* axis remained almost unchanged between 123 and 333 K and then began contracting by 6.29% from 333 to 438 K. Consequently, volumetric expansion of 6.10% was observed (Figs 2 and 3a).

This phase transition from the orthorhombic phase to the tetragonal phase was thermally irreversible; the tetragonal

structure of compound **1t** was maintained upon cooling from 438 to 123 K (Fig. 3; Supplementary Fig. 6). Variable-temperature powder X-ray diffraction (Supplementary Fig. 7) and differential scanning calorimetry (DSC; Fig. 4b) experiments confirmed the thermal irreversibility of the phase transition. In the powder X-ray diffraction patterns, with increasing temperature, the first two peaks at 6.54° and 7.57° at 303 K (indexed as [0 2 0] and [2 0 0], respectively, in the orthorhombic phase) gradually approach each other and finally irreversibly merge into a single peak at 6.89° (indexed as [1 1 0] in the tetragonal phase). The DSC curve of compound **1o** shows a broad exothermic peak at ca. 445 K during heating (estimated $\Delta H$ of ca. $-1.53$ kJ mol$^{-1}$), whereas no heat-flow peak was observed upon cooling, suggesting that an irreversible chemical or physical change occurred in the solid state. To completely preclude the existence of a large hysteresis for the phase transition, compound **1t** was cooled to 5 K with liquid He on a SQUID magnetometer. The powder X-ray diffraction measurement at 303 K for the pre-cooled sample revealed that the tetragonal phase was maintained, which further confirmed the thermal irreversibility of the phase transition (Supplementary Fig. 8).

To confirm the phase transition is thermally induced rather induced by guest loss, we synthesized the isostructural compound [(H$_4$BPTC)·(BPE)$_2$]·0.33EtOH (**1o**·0.33EtOH), whose formula was determined by TG analysis (Supplementary Fig. 5). Powder X-ray diffraction analysis of the desolvated sample revealed an identical phase transition temperature (Supplementary Fig. 9).

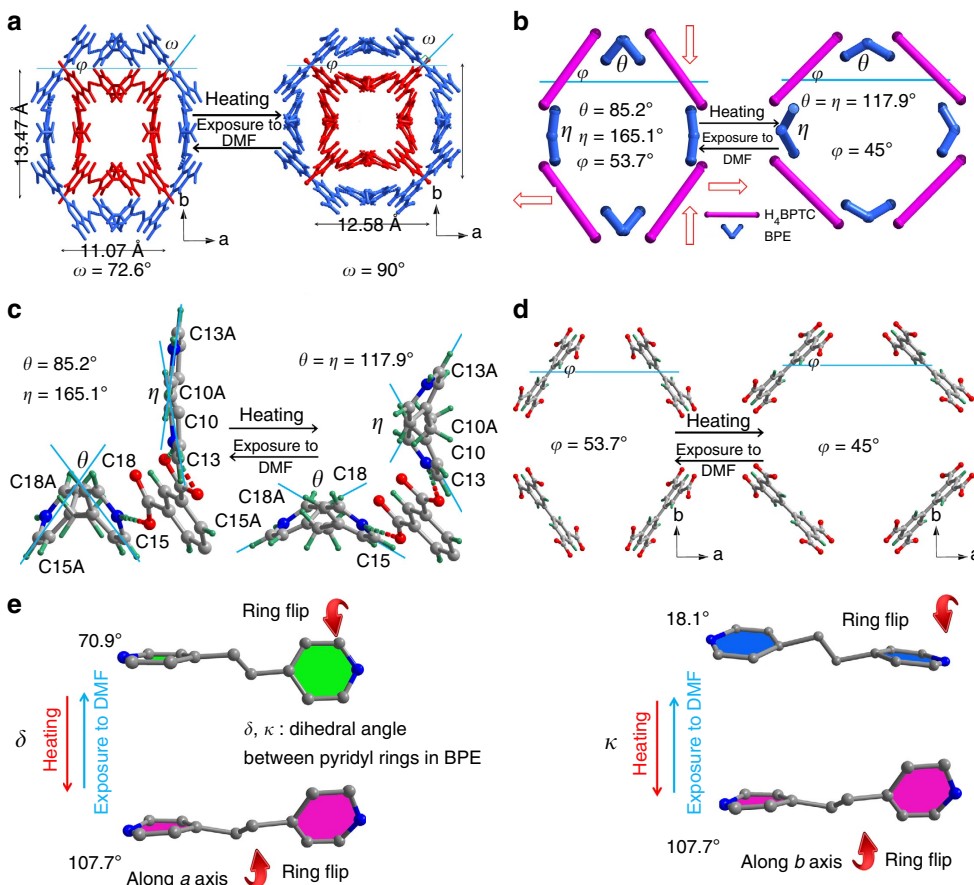

**Figure 2 | Structural evolution at the molecular level from compound 1o at 123 K to compound 1t at 438 K.** (**a**) The interlocking angle $\omega$ changing from 72.6° to 90°; (**b**) scheme showing the structural changes; (**c**) structures showing BPE acting as a molecular spring along the *a* and *b* axes via pyridyl ring-flipping; (**d**) structures showing H$_4$BPTC lying down toward the *a* axis as a molecular rotor; and (**e**) scheme illustrating the dihedral angles $\delta$ and $\kappa$ between pyridyl rings in BPE changing significantly because of pyridyl ring-flipping.

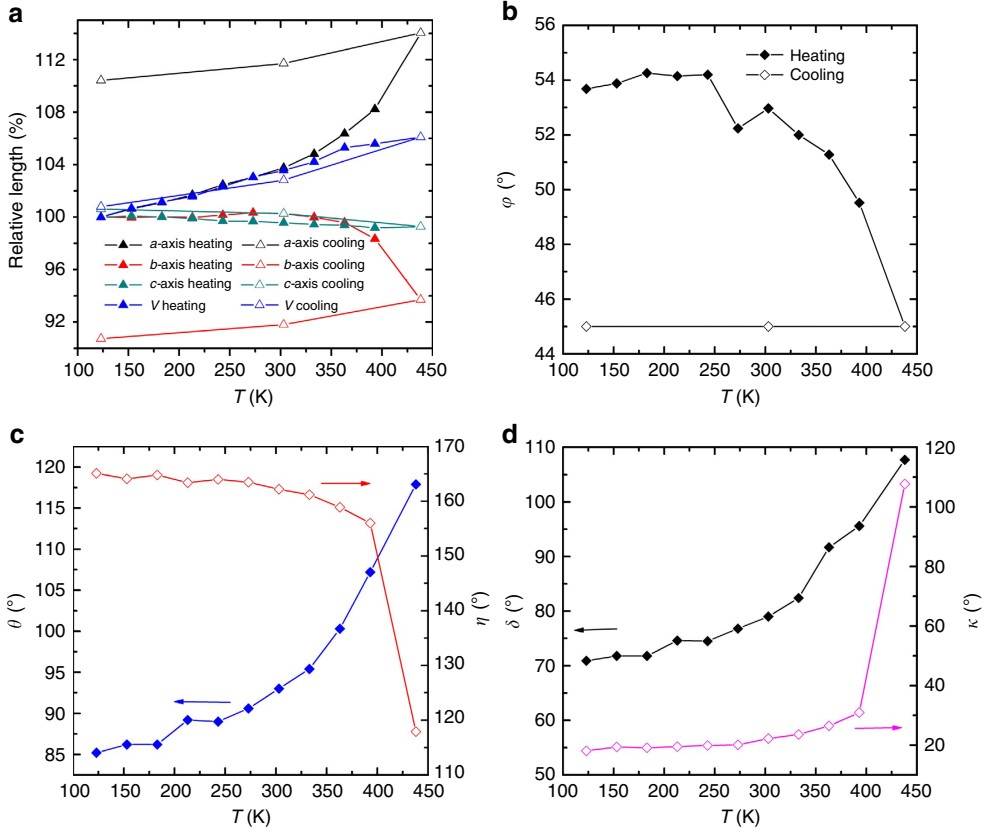

**Figure 3 | Evolution of structural parameters.** (**a**) Evolutions of temperature-dependent unit-cell parameters; all parameters are normalized by those determined at 123 K; the unit cell of compound **1t** has been transferred to that of compound **1o** for direct comparison (that is, $a_o = \sqrt{2} \times a_t$, $b_o = \sqrt{2} \times b_t$, $c_o = c_t$, $V_o' = 2 \times V_t$); evolution of temperature-dependent (**b**) angle $\varphi$, (**c**) angles $\theta$ and $\eta$, and (**d**) angles $\delta$ and $\kappa$. During the phase transition, angles $\eta$ and $\kappa$ undergo a dramatic change, implying that the pyridyl rings in BPE along the $b$ axis flip drastically.

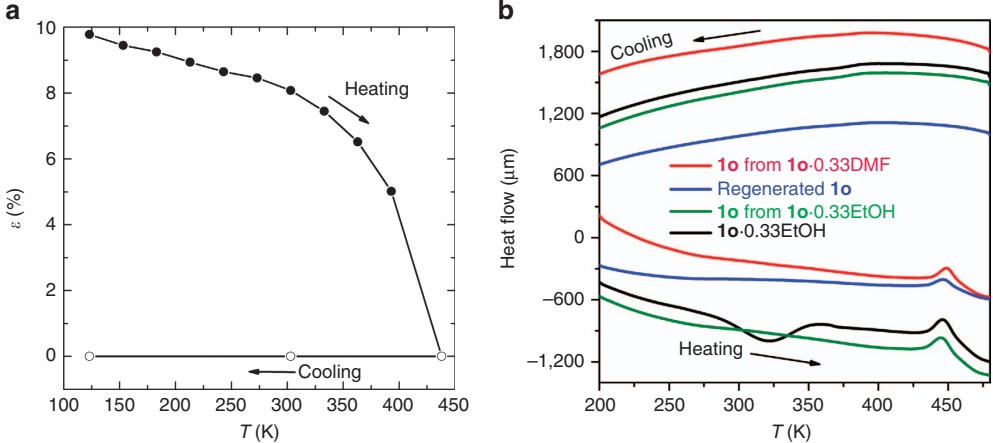

**Figure 4 | Spontaneous strain variation and DSC curves.** (**a**) Thermal variation of the spontaneous strain $\varepsilon$ as defined in equation (1); and (**b**) DSC curves showing the thermally irreversible phase transition from compound **1o** to compound **1t**.

In addition, the heating branch of the DSC curve of compound **1o**·0.33EtOH shows a broad endothermic peak at ca. 323 K and an exothermic peak at ca. 445 K, which correspond to the loss of solvent and to the phase transition, respectively; by contrast, the DSC curve for the desolvated sample shows only the exothermic peak at ca. 445 K. These results clearly indicate the transition from the orthorhombic phase to the tetragonal phase is thermally induced, not relying on guest molecules (Fig. 4b).

The single-crystal-to-single-crystal phase transition is accompanied by mechanical motion of the crystal. The molecular mobility at the sub-nanometre scale within the crystal is translated into a macroscopic crystal shape change because of the concerted and collective motion of the molecules in the whole crystal. Photomicrographs of a single crystal recorded at 123 and 453 K show that the length of the crystal along the $b$ axis contracted ~6% upon heating, whereas the shape of the single

crystal remained almost unchanged upon cooling to 123 K (Supplementary Fig. 10). The transition from $Ccca$ to $P4_2/nbc$ is a transition from a ferroelastic phase to a paraelastic phase, with a total symmetry increase from eight symmetric elements ($E$, three $C_2$, $i$ and three $\sigma$) to sixteen ($E$, two $C_4$, five $C_2$, $i$, two $S_4$ and five $\sigma$) and Aizu notation of $4/mmmFmmm$ (ref. 23). As a ferroic property, ferroelasticity is defined by a strain–stress hysteresis. Because accurately measuring ferroelastic hysteresis is difficult, a material is customarily referred to as 'ferroelastic' if a phase transition that may generate ferroelasticity occurs. Theoretically, such phase transitions should belong to the 94 species deduced by Aizu[23–25].

**Thermoelasticity**. Compound **1o** exhibits superior reversible thermoelasticity withstanding large biaxial strain below the phase transition temperature (Supplementary Table 3; Supplementary Fig. 11). The variable-temperature unit-cell determinations on compound **1o** during cooling from 408 to 123 K and then reheating to 408 K, revealed that compound **1o** exhibits reversible colossal positive thermal expansion (PTE) along the $a$ axis and negative thermal expansion (NTE) along the $b$ axis. In the temperature range from 123 to 408 K, the $a$ axis expands 11.2%, with a colossal expansion coefficient $\alpha_a$ in the range $125 \times 10^{-6} < \alpha_a < 394 \times 10^{-6}$; the $b$-axis contracts 4.0% from 333 to 408 K, with an expansion coefficient $\alpha_b$ in the range $-171 \times 10^{-6} < \alpha_b < -469 \times 10^{-6}$ (Supplementary Table 4; Supplementary Fig. 11). In particular, in the temperature range from 333 to 408 K, the $a$ and $b$ axes expand and contract with giant expansion coefficients $\alpha_a$ and $\alpha_b$ in the range $450 \times 10^{-6} < \alpha_a < 814 \times 10^{-6}$ and $-171 \times 10^{-6} < \alpha_b < -469 \times 10^{-6}$, respectively (Supplementary Table 5). The giant PTE and NTE of compound **1o** in the range $333-408$ K are much larger than those for other known crystalline molecular crystals and MOFs (Supplementary Table 6)[26–31]. The photomicrographs of a single crystal recorded at 123 and 408 K show that the length of the single crystal ($b$ axis) contracts ∼4% from 123 to 408 K. Moreover, such significant thermally driven crystal shape change could be repeatedly observed up to ten cycles without any observable cracks in the crystal (Supplementary Fig. 12). However, after the phase transition, the framework breathing in the space group $P4_2/nbc$ is symmetry forbidden, leading to a substantial increase in rigidity. The values of strain $\varepsilon = [(b - a)/(a + b)]$ (1) (for a transition between orthorhombic and tetragonal phases)[32] determined from our variable-temperature crystallographic data are shown in Fig. 4a.

**Thermally induced structural evolution**. A systematic comparison of the variable-temperature crystal structures enabled us to rationalize the mechanism of the unusual thermal expansion at the molecular level. The structures of both H₄BPTC and BPE are dynamic under changing temperature (Supplementary Table 7). The H₄BPTC gradually lies down toward the $a$ axis, with a decrease in angle $\varphi$ from 53.7° at 123 K to 49.5° at 393 K, and then a sudden decrease to 45° at 438 K, leading to an increase in the interlocking angle $\omega$ from 72.6° at 123 K to 90° at 438 K (Figs 2 and 3). BPE functions as a molecular spring in the framework via pyridyl ring-flipping. The pyridyl rings along the $a$ axis stretch, with the torsion angle $\theta$ (C15-C18-C18A-C15A) gradually increasing from 85.2° at 123 K to 117.9° at 438 K. The rings along the $b$ axis shrink, with the torsion angle $\eta$ (C13-C10-C13A-C10A) slightly decreasing from 165.1° at 123 K to 156.0° at 393 K and then drastically to 117.9° at 438 K for the phase transition (Figs 2 and 3). Both structural evolutions contribute positively to the colossal PTE of the $a$ axis and NTE of the $b$ axis. During the structural evolution, the conformation of

H₄BPTC remains almost static, whereas the conformations of BPE change substantially. For the BPE along the $a$ axis, the dihedral angle $\delta$ between two pyridyl rings gradually increases from 70.9° at 123 K to 107.7° at 438 K. For the BPE along the $b$ axis, the dihedral angle $\kappa$ slightly increases from 18.1° at 123 K to 30.9° at 393 K and then drastically to 107.7° at 438 K for the pyridyl ring-flipping during the phase transition (Figs 2 and 3). As a result, two crystallographically independent BPE halves in compound **1o** become symmetry related in compound **1t**. Compound **1t** at 123 K has a void space of 508 Å³ per unit cell, which is also ∼13% of the unit-cell volume. However, the square pore shape differs from the rectangular shape in compound **1o**, and the channel size changes from $3.2 \times 4.0$ Å² in compound **1o** at 123 K to $3.7 \times 3.7$ Å² in compound **1t** at 123 K.

**Guest-induced phase transition**. Compound **1t** recovers to the original orthorhombic phase after being exposed to DMF vapour, as indicated by the powder X-ray diffraction (Supplementary Fig. 6) and DSC (Fig. 4b inset) results, which indicated the induction of a shape-memory effect. TG analysis of the sample exposed for 4 days indicated that ∼6.8 wt% DMF was adsorbed. In the powder X-ray diffraction patterns, the first peak at 7.05° at 303 K split into two peaks at 6.56° and 7.58°. This transition is attributed to the host–guest interaction generating stress that causes the tetragonal phase to deform via a ferroelastic transition. However, the highly disordered solvent in the pores cannot be crystallographically located, precluding quantum chemical calculations of the energy profile of the guest-induced ferroelastic transition. The DSC trace of the regenerated compound **1o** also shows a broad exothermic peak at ca. 445 K upon heating.

**Sorption properties**. Two interconvertible empty phases with different pore shapes were successfully isolated, which enabled us to study their respective adsorption properties. Compound **1o** completely rejects N₂ at 77 K, which is attributed to the dimensions of its pore ($3.2 \times 4.0$ Å²) not matching the size of N₂ molecules ($d = 3.64$ Å), preventing the diffusion of N₂ into the pores. The pore dimensions of compound **1t** ($3.7 \times 3.7$ Å²) are compatible with N₂ molecules; consequently, N₂ is weakly adsorbed into compound **1t**, with a Brunauer–Emmett–Teller (BET) surface area of ca. 33 m² g⁻¹ and an uptake of 31 cm³ g⁻¹ at $P/P_0 = 0.93$, implying a pore volume of ca. 0.048 cm³ g⁻¹ (Supplementary Fig. 13). The high uptake at $P/P_0$ of ∼1.0 may be attributed to the intercrystal porosity formed by the packing of non-uniform crystals. Unexpectedly, unlike other HOFs, compounds **1o** and **1t** both exclude CO₂ at 196 K (Supplementary Fig. 14)[33,34].

**Discussion**

To understand the driving force of the thermally irreversible transition from compound **1o** to compound **1t**, we performed quantum chemical calculations of the energy profile of the transition (Fig. 5). Because the main effect of the thermal treatment of compound **1o** is to reduce the spontaneous strain $\varepsilon$ defined in equation (1), we calculated the relative single-point energy of the structure of compound **1o** at various values of strain $\varepsilon$ in the structural evolution and compared the values with that of compound **1t**, which was set to be 0. Our calculations indicated that compound **1o** becomes unstable when $\varepsilon$ approaches 0, where compound **1o** tends to transition to compound **1t**. DSC measurements revealed an estimated $\Delta H$ of ca. $-1.53$ kJ mol⁻¹, indicating that above $E$ of ca. 1.53 kJ mol⁻¹, compound **1o** is unstable and quickly transitions to the energetically preferred compound **1t** (Fig. 5, inset). Compound **1o** is energetically preferred at lower temperatures, whereas compound **1t** is

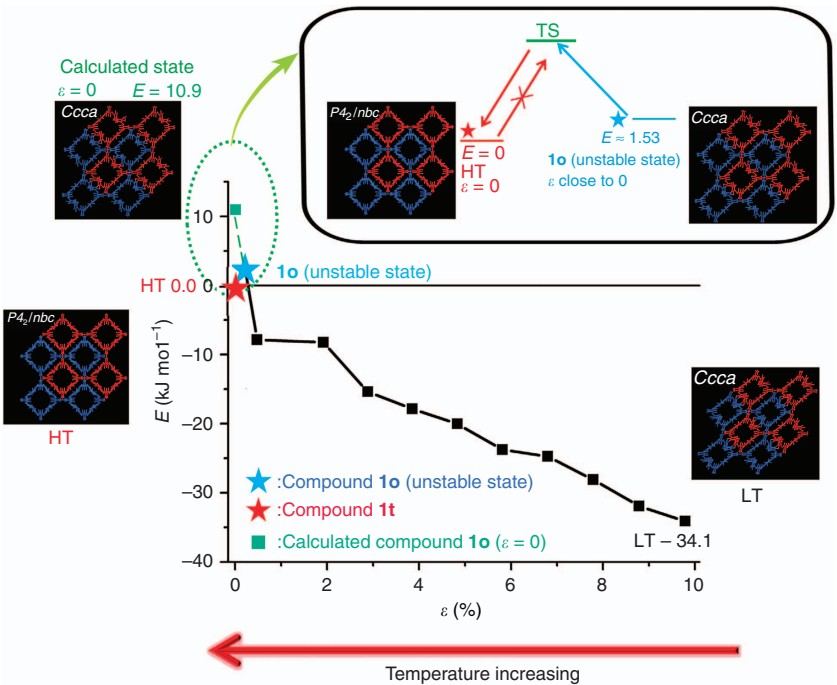

**Figure 5 | Calculated energy profile.** Calculated energy profile for compound **1o** as a function of strain ε (inset: energy diagram for the thermally irreversible transition from compound **1o** to compound **1t**). The energy for compound **1t** is set to be 0. The structural figures represent optimized structures (by calculation), and the ε value for compound **1o** (that is, the unstable state) in the figure is set to be 0.1%; red and blue frameworks represent the pair of sextuple interlocked frameworks with opposite handedness. At lower temperatures, compound **1o** is energetically preferred and exists with a large ε value. As the temperature increases, the ε value decreases and the potential energy increases. When ε approaches 0, compound **1t** is energetically preferred and compound **1o** transitions to compound **1t**. The reverse transition (that is, from compound **1t** to compound **1o**) requires the initial activation of a symmetry-breaking process that involves overcoming an energy barrier. However, upon cooling, no heat can be gained to overcome such an energy barrier; therefore, the transition from compound **1t** to compound **1o** cannot occur upon cooling, resulting in stabilization of metastable compound **1t** at lower temperatures. LT, HT and TS represent low-temperature phase, high-temperature phase and transition state, respectively.

energetically preferred at higher temperatures; therefore, the structural strain induced by guest incorporation cannot be removed by desolvation under vacuum but can be removed by thermal treatment via a reverse ferroelastic transition (from compound **1o** to compound **1t**). However, the reverse transition (from compound **1t** to compound **1o**) requires the initial activation of a symmetry-breaking process that involves overcoming an energy barrier; only after this energy barrier has been overcome can the structure transition to the orthorhombic phase and then relax to the lower-energy states with large ε values. Upon cooling, no heat can be gained to overcome such an energy barrier, making the transition from compound **1o** to compound **1t** thermally irreversible. However, the ferroelastic transition (from compound **1t** to compound **1o** · 0.33DMF) can be initiated by solvent incorporation, leading to the observed shape-memory effect, which is similar to that observed in small crystals of $[Cu_2(bdc)_2(bpy)]_n$ (bdc = 1,4-benzenedicarboxylate, bpy = 4,4′-bipyridine)[14]. In small crystals of $[Cu_2(bdc)_2(bpy)]_n$, a metastable open dried phase was stabilized at lower temperatures by crystal downsizing. In our SOF, both the superior thermoelasticity of compound **1o** and the energy barrier that involves symmetry breaking are important to the shape-memory effect. The superior thermoelasticity makes compound **1o** unstable at higher temperatures; thus, it transitions to compound **1t**. The energy barrier stabilizes **1t** as a metastable phase at lower temperatures. Both effects are essential to the shape-memory behaviour.

In conclusion, we demonstrated that superior thermoelasticity and shape-memory behaviour can be achieved in a highly flexible supramolecular organic framework. This observation provides an alternative approach to control the porosity other than by guest removal or incorporation, which may be applicable to other flexible porous structures. The superior thermoelasticity and the elucidated mechanism of the thermally irreversible phase transition may be useful for designing temperature-controlled smart materials. These shape-memory single crystals offer great practical potential for actuators, intelligent devices, and artificial muscles.

## Methods

**Materials.** All reagents except H$_4$BPTC were purchased commercially and used without purification.

**Synthesis.** H$_4$BPTC: H$_4$BPTC was synthesized by a modified literature procedure[35]. 3,3′,5,5′-Tetramethylbiphenyl (1.0 g, 0.0047 mol) was oxidized with KMnO$_4$ (6.5 g, 0.112 mol) in *tert*-butanol/water (v/v = 1:1; 50 ml) containing NaOH (0.4 g, 0.01 mol). Yield: 1.12 g (72.2%). Anal. calcd (found) for C$_{16}$O$_8$H$_{10}$: C, 58.19% (58.10%); H, 3.05% (3.09%).

[(H$_4$BPTC) · (BPE)$_2$] · 0.33DMF (**1o** · 0.33DMF): a mixture of H$_4$BPTC (0.033 g, 0.1 mmol) and 1,2-bis(4-pyridyl)ethane (0.036 g, 0.2 mmol) in DMF (5 ml) was sealed in a 20 ml glass vial at 120 °C for 2 days, and then cooled to room temperature. The clear solution was allowed to stand at room temperature for ∼1 week. Colourless needle-shaped crystals of compound **1o** · 0.33DMF were filtrated and air dried (yield: 70.5%).

[(H$_4$BPTC) · (BPE)$_2$] · 0.33EtOH (**1o** · 0.33EtOH): a mixture of H$_4$BPTC (0.066 g, 0.2 mmol) and 1,2-bis(4-pyridyl)ethane (0.072 g, 0.4 mmol) in EtOH (8 ml) was sealed in a 20 ml teflon-lined autoclave at 160 °C for 6 h and then cooled to room temperature. Colourless needle-shaped crystals of compound **1o** · 0.33EtOH were filtrated and air dried (yield: 78.6%). The formula was determined by thermogravimetric analysis (Supplementary Fig. 5).

Compound **1o** was obtained by desolvation of compound **1o** · 0.33DMF or **1o** · 0.33EtOH at 60 °C under vacuum for 12 h. After desolvation, the sample was kept under vacuum and used for characterization immediately. Compound **1t** was

obtained by heating compound **1o** to 180 °C. Regenerated compound **1o** was obtained by exposing compound **1t** to DMF vapour for 4 days, followed by re-desolvation.

**Measurements.** Elemental analyses were performed using a Vario EL elemental analyzer, after drying the samples with $P_2O_5$ under vacuum for 12 h. IR (KBr pellet) spectra were recorded in the range of 400–4,000 $cm^{-1}$ on a JASCO FT/IR-600 Plus spectrometer. Thermogravimetric analyses were performed using a TG/DTA6300 system at a rate of 5 °C min$^{-1}$. Differential scanning calorimetry (DSC) measurements were performed using a XDSC-7000 system at a rate of 5 °C min$^{-1}$ powder X-ray diffraction patterns were obtained on a Rigaku 2100 diffractometer with Cu $K_\alpha$ radiation in the flat plate geometry. The temperature was increased at a rate of 20 °C min$^{-1}$ and kept for 5 min for measurement at each targeted temperature. $^1H$ NMR spectra were recorded on a Hitachi 400 MHz spectrometer.

**Single-crystal X-ray diffraction.** Diffraction data were collected on a Rigaku CCD diffractometer with Cu $K_\alpha$ radiation. The measurement temperature was changed at a rate of 10 °C min$^{-1}$ and maintained for 5 min before measurement at each targeted temperature. The structures were solved by the direct method and refined by the full-matrix least-squares technique on $F^2$ using the SHELX programme. Hydrogen atoms were geometrically generated and refined by a riding model. Variable-temperature unit cell parameters for compound **1o** were determined by collecting 180 diffraction images at each temperature. The SQUEEZE method was applied to remove the highly diffused electron density in all the structures. CCDC 1423054 and CCDC 1423057–1423069 contain the supplementary crystallographic data for this paper, the details are listed in Supplementary Tables 1 and 2.

**Single-crystal pictures.** Single-crystal pictures were taken with a camera equipped on a Rigaku CCD diffractometer. The temperature was increased at a rate of 20 °C min$^{-1}$ and maintained for 5 min at each targeted temperature.

**Adsorption measurements.** Gas adsorption measurements were performed on an ASAP 2020. Before the adsorption measurement, ∼100 mg of sample was placed in the sample cartridge and outgassed under vacuum. After outgassing, the system was maintained under vacuum at the targeted temperature. For compound **1o**, the sample was activated at 120 °C under vacuum for 24 h. Compound **1t** used for the sorption measurement was obtained by heating compound **1o** at 180 °C for 5 min.

**Computational methods.** All calculations were performed with the DMol3 programme[36,37] in Material Studio (Accelrys, Inc.). The Perdew–Burke–Ernzerhof (PBE) generalized gradient functional was employed for the exchange-correlation energy. The wave functions were expanded in terms of numerical basis sets. We employed the DND basis set (double numerical basis set with d-type polarization functions) for geometry optimization. The Brillouin zone was sampled with a $(1 \times 1 \times 1)$ Monkhorst–Pack[38] mesh of k-points. To reasonably describe weak interactions between organic molecules, we used the dispersion correction method developed by Tkatchenko and Scheffler[39]. In addition, atomic coordinates were optimized as a function of the strain parameter $\varepsilon = [(b-a)/(a+b)]$, where $a$ and $b$ are unit cell parameters.

**Data availability.** The X-ray crystallographic coordinates for structures reported in this article have been deposited at the Cambridge Crystallographic Data Centre (CCDC), under deposition numbers CCDC 1423054, CCDC 1423057–1423069. These data can be obtained free of charge from The Cambridge Crystallographic Data Centre via www.ccdc.cam.ac.uk/data_request/cif.

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

## Acknowledgements

Y.H. thanks the Japan Society for the Promotion of Science for a postdoctoral fellowship, (No. P13033). This work was supported by Grants-in-Aid (Nos. 26104528, 15H01018, 25288029, 24109014, 24550190, and 15K13710) from Japan Society for the Promotion of Science (JSPS) and the Ministry of Education, Culture, Sports, Science and Technology of Japan (MEXT), the MEXT Projects of 'Integrated Research on Chemical Synthesis', 'Elements Strategy Initiative to Form Core Research Center', and CREST, Japan Science and Technology Agency(JST) 'Innovative Catalyst'.

## Author contributions

Y.-G.H. and O.S. designed this study and wrote the manuscript. Y.-G.H conducted the experiments. S.-Q.S., Z.-S.Y., G.-L.L., S.-Q.W., K.A., S. Kanegawa and S. Kang contributed by measuring the crystal deformation and diffraction. M.-Y.W. and M.-C.H. performed the sorption measurements. Y.S., T.K. and K.Y. performed the calculations and wrote the related discussion. All authors discussed the results and commented on the manuscript.

## Additional information

**Competing financial interests:** The authors declare no competing financial interests.

