## [Peer Review File · Nature Communications]

Reviewer #2 (Remarks to the Author):

The manuscript has been improved through the revision. This reviewer is supportive of the acceptance.

Reviewer #3 (Remarks to the Author):

Review for revised manuscript

The revised manuscript for this work is considerably improved as related to both science and readability. In addition to the content with the first submission, this manuscript now reports newly developed experiments that offer further glimpses to the intrigue of this new HOF. The recently added areas include such topics as N₂ and CO₂ isotherms, variable temperature pXRD data, and greater crystallographic detail of the phase transition. The study remains most interesting in terms of its contribution as a HOF material capable of an irreversible temperature initiated orthorhombic to tetragonal phase change. This reported thermoelastic behavior is highly anisotropic and represents a first for HOF materials. The question has been raised by the other reviewers whether this work meets the standard for Nature Chemistry. It is quite true the areas of temperature-dependent phase transitions and materials that exhibit gas adsorption/desorption are well covered in the literature. The originality of this work rests with the shape-switchable nature of this novel HOF. Thinking forward, the functional properties of this material potentially opens a new direction for others to follow that could provide answers to current/future science challenges. In the opinion of this reviewer, Nature Chemistry (or a move to Nature Comm) still seems like an appropriate fit for this manuscript. Moving this work to a more specialized journal would dilute its potential impact to the community.

Reviewer #4 (Remarks to the Author):

Most of the comment of the referees have been taken into account any extensive complementary experiments have been carried out to shed some light of the irreversibility of the phase transition. This article is thus suitable for publication, although I am still not convinced, because of the lack of true novelty or performance, that it deserves publication in such a high impact journal as Nature Communications.

Remark : in Figure S9 (temperature dependent PXR of 1o.EtOH), the first diffraction peak on the -40°C diagram seems to split in two peaks, is it true? If so, is this the signature of the 1o form ?

Response to reviewer 2

Reviewer Comments:

Reviewer #2 (Remarks to the Author):

The manuscript has been improved through the revision. This reviewer is supportive of the acceptance.

Reply:

We would like to thank the referee for the careful review and valuable comments.

Response to reviewer 3

Reviewer #3 (Remarks to the Author):

Review for revised manuscript

The revised manuscript for this work is considerably improved as related to both science and readability. In addition to the content with the first submission, this manuscript now reports newly developed experiments that offer further glimpses to the intrigue of this new HOF. The recently added areas include such topics as N₂ and CO₂ isotherms, variable temperature pxrd data, and greater crystallographic detail of the phase transition. The study remains most interesting in terms of its contribution as a HOF material capable of an irreversible temperature initiated orthorhombic to tetragonal phase change. This reported thermoelastic behavior is highly anisotropic and represents a first for HOF materials. The question has been raised by the other reviewers whether this work meets the standard for Nature Chemistry. It is quite true the areas of temperature-dependent phase transitions and materials that exhibit gas adsorption/desorption are well covered in the literature. The originality of this work rests with the shape-switchable nature of this novel HOF. Thinking forward, the functional properties of this material potentially opens a new direction for others to follow that could provide answers to current/future science challenges. In the opinion of this reviewer, Nature Chemistry (or a move to Nature Comm) still seems like an appropriate fit for this manuscript. Moving this work to a more specialized journal would dilute its potential impact to the community.

Reply:

We would like to thank the referee for the careful review and valuable comments.

Response to reviewer 4

Reviewer #4 (Remarks to the Author):

Most of the comment of the referees have been taken into account any extensive complementary experiments have been carried out to shed some light of the irreversibility of the phase transition. This article is thus suitable for publication, although I am still not convinced, because of the lack of true novelty or performance, that it deserves publication in such a high impact journal as Nature Communications.

Remark : in Figure S9 (temperature dependent PXRD of 1o.EtOH), the first diffraction peak on the -40°C diagram seems to split in two peaks, is it true? If so, is this the signature of the 1o form ?

Reply:

We would like to thank the referee for the careful review and valuable comments. In our first revision, the top of the peak was masked by the simulated PXRD pattern of compound **1t** at 303 K, which seemed to split. In reality, it does not. The peak can be seen more clearly in the following supplementary figure, wherein only two diagrams (-40°C and simulated **1t** at 303 K) are shown.

Supplementary Figure. PXRD pattern for compound **1t** at -40°C .

In our second revision, the two PXRD patterns (-40°C and simulated **1t** at 303 K) are separately shown in Supplementary Figure 9, wherein these two PXRD patterns do not overlap.